# Mucosal IFNλ1 mRNA-based immunomodulation effectively reduces SARS-CoV-2 induced mortality in mice

Anna Macht[1,4], Yiqi Huang [ID][2,4], Line S Reinert [ID][3,4], Vincent Grass [ID][2], Kristin Lohmer[1], Elke Tatjana Aristizabal Prada[1], Eveline Babel[1], Alexandra Semmler[1], Wen Zhang[1], Andrea Wegner[1], Eva Lichtenegger-Hartl[1], Sonja Haas[1], Günther Hasenpusch[1], Steffen Meyer[1], Søren R Paludan [ID][3,4], Andreas Pichlmair [ID][2,4], Carsten Rudolph[1,4] & Thomas Langenickel [ID][1,4 ✉]

## Abstract

**RNA vaccines elicit protective immunity against SARS-CoV-2, but the use of mRNA as an antiviral immunotherapeutic is unexplored. Here, we investigate the activity of lipidoid nanoparticle (LNP)-formulated mRNA encoding human IFNλ1 (ETH47), which is a critical driver of innate immunity at mucosal surfaces protecting from viral infections. IFNλ1 mRNA administration promotes dose-dependent protein translation, induction of interferon-stimulated genes without relevant signs of unspecific immune stimulation, and dose-dependent inhibition of SARS-CoV-2 replication in vitro. Pulmonary administration of IFNλ1 mRNA in mice results in a potent reduction of virus load, virus-induced body weight loss and significantly increased survival. These data support the development of inhaled administration of IFNλ1 mRNA as a potential prophylactic option for individuals exposed to SARS-CoV-2 or at risk suffering from COVID-19. Based on the broad antiviral activity of IFNλ1 regardless of virus or variant, this approach might also be utilized for other respiratory viral infections or pandemic preparedness.**

**Keywords** Immunology; Infectious Diseases; Interferon; Nucleic-acid Therapeutics; Viral Infection
**Subject Categories** Immunology; Methods & Resources; Microbiology, Virology & Host Pathogen Interaction

## Introduction

Several human pathogens, including SARS-CoV-2, influenza virus or respiratory syncytial virus can cause severe respiratory diseases (Flerlage et al, 2021; Paludan and Mogensen, 2022). Interferons (IFNs) are key cytokines orchestrating cell intrinsic antiviral immune responses (Haller et al, 2006; Hoffmann et al, 2015) through induction of interferon-stimulated genes (ISGs) with antiviral and immunomodulatory functions (Pervolaraki et al, 2018; Syedbasha and Egli, 2017). IFNλs are the first and most highly expressed IFNs produced by epithelial cells at the site of virus entry following infection (Mordstein et al, 2010; Ye et al, 2019). IFNλs have potent antiviral functions and suppress initial viral spread without induction of broad inflammatory responses (Davidson et al, 2016), due to the restricted expression of IFNλ receptor (IFNLR) to epithelial cells and a very limited subset of inflammatory cell types (Sheppard et al, 2003; Sommereyns et al, 2008; Uze et al, 2007).

In contrast, the receptor for IFNs other than IFNλ is broadly expressed on a large spectrum of cells, enhancing viral resistance but also inducing aberrant inflammatory responses that can cause immunopathology (Sleijfer et al, 2005). Among lambda interferons, IFNλ1 is preferred as IFNλ2/3 responses present with delayed kinetics (Osterlund et al, 2007). Based on its broad antiviral activity, local delivery to the upper and lower respiratory tract of human IFNλ1 (hIFNλ1) could be utilized for prophylaxis and/or treatment of respiratory virus infections independent of virus type and variant (Andreakos and Tsiodras, 2020; Galani et al, 2017).

Utilization of mRNA encoding hIFNλ1 may provide additional advantages over recombinant IFN therapy as mRNA is processed by cellular translation machinery to produce endogenous protein, minimizing the risk of inducing anti-drug antibodies. Moreover, the half-life of endogenously expressed protein is longer compared to exogenously administered recombinant IFN (Eichinger et al, 2017), thus potentially allowing longer-lasting protective effects. One additional advantage of mRNA over recombinant protein therapy is that IFN translated from mRNA can be secreted to the basolateral side of the epithelium, where IFNLRs are expressed (Stanifer et al, 2020). Local administration of mRNA can be achieved with nasal application via nasal spray or pulmonary application using a nebulizer, which enables administration of

[1]ETHRIS GmbH, Planegg, Germany. [2]Institute of Virology, Technical University of Munich, Munich, Germany. [3]Department of Biomedicine, Aarhus University, Aarhus, Denmark. [4]These authors contributed equally: Anna Macht, Yiqi Huang, Line S Reinert, Søren R Paludan, Andreas Pichlmair, Carsten Rudolph, Thomas Langenickel. ✉E-mail: langenickel@ethris.com

lower doses of drug product with high bioavailability at its site of pharmacological activity (Fig. 1).

## Results and discussion

### Treatment of A549 lung cells with ETH47 leads to dose-dependent translation of target protein and target gene induction without relevant cytokine increase

Pharmacology of ETH47 (LNP-formulated IFNλ1-encoding mRNA) was investigated in A549 cells, a human alveolar basal epithelial cell line. Dose-dependent hIFNλ1 protein secretion was observed as early as 3 h after ETH47 administration. IFNλ1 levels peaked at 24 h and persisted for up to 72 h in the higher dose groups (Fig. 2A). IFNλ1 concentration in samples treated with Control-LNP (double dose of ETH47) was measured at 24 h timepoint and values were below limit of quantification (see Source Data). ETH47 administration also led to a dose-dependent induction of the ISGs IFIT1 and ISG15, which peaked at 24 h (Fig. 2B,C). Another ISG, OAS3, was also induced, but with peak induction observed at 72 h (Fig. 2D). All ISGs remained activated for at least 72 h (last measurement). No target gene induction was observed in a dose-range finding experiment using LNP-formulated non-translatable Control-mRNA at dose levels exceeding those of ETH47, confirming that observed ISG induction was target-related (Fig. EV1). Minimal and mostly transient induction of IL-8 and IL-6 transcription was observed at the highest dose level of ETH47, while MCP-1 mRNA transcription was not induced (Fig. 2E–G). These results suggest that ETH47 treatment does not induce an inflammatory response in this setting, although the relevance of this observation would need to be established in more complex immunological models.

We further tested translation of ETH47 in primary human air–liquid interface (ALI) cultures, a model representative of human respiratory tissue, and compared it to apical treatment with recombinant hIFNλ1. ETH47 treatment resulted in larger hIFNλ1 concentrations at the basal side of the epithelium compared to the apical side and to administration of recombinant hIFNλ1 (basal/apical ratio of 6 with ETH47 vs. 3 with recombinant protein; (Fig. 2H). Larger concentrations on the basolateral side could be beneficial in vivo, as IFNLR1 is expressed on both apical

and basolateral sides of epithelial cells (Fox et al, 2015), and basolateral delivery would prevent virus spread into the tissue.

Above in vitro results show that ETH47 leads to (1) dose-dependent target protein translation, which is more pronounced at the basolateral side of ALI cultures, (2) activation of interferon target genes with antiviral properties, which is more sustained than IFNλ1 protein translation, and (3) generally low (if not negligible) induction of potentially undesired cytokine transcription prior to achieving plateau of IFNλ1 protein translation and target gene induction.

### Administration of ETH47 mRNA results in potent inhibition of SARS-CoV-2 virus replication in vitro

The effect of ETH47 mRNA compared to Control-mRNA and recombinant hIFNλ1 on SARS-CoV-2 replication was tested using A549 cells stably expressing the ACE2 receptor (A549-ACE2). SARS-CoV-2 infection did not influence dose-dependent generation of hIFNλ1 following administration of ETH47 mRNA (Fig. 3A), and administration of Control-mRNA did not result in measurable IFNλ1 protein induction. Further, ETH47 mRNA resulted in potent dose-dependent reduction of SARS-CoV-2 virus load by up to 100-fold compared to Control-mRNA (Fig. 3B). Virus replication was inhibited at the lowest tested dose and maximally inhibited at the mid-dose level. No additional inhibition of virus replication was observed at higher doses despite a further increase in secreted IFNλ1 protein. Control-mRNA did not inhibit virus replication (Fig. 3B), indicating that non-translatable mRNA does not render cell cultures resistant to viral replication. Notably, administration of recombinant hIFNλ1 resulted in approximately fourfold less reduction of virus load compared to ETH47 at a dose that results in comparable IFNλ1 concentrations in cell culture medium (see asterisk in Fig. 3B). Collectively, this demonstrates that ETH47 mRNA treatment results in potent antiviral immunity against SARS-CoV-2 in A549-ACE2 cells.

### Single nasal administration of ETH47 results in target gene induction without undesired immune activation in mice

In vivo efficacy of ETH47 was tested in C57BL/6 mice. Aiming to gain a broader view regarding protein expression and target gene engagement at several timepoints with three ascending doses, animal number per condition was kept low. Results showed that single nasal administration of ETH47 at doses of 0.025, 0.125 and 0.35 mg/kg bw (body weight) leads to dose-dependent pulmonary hIFNλ1 translation at 5 h post treatment. At 24 h, no hIFNλ1 could be detected (Fig. 4A), indicating that ETH47 leads to transient translation of hIFNλ1. ETH47 also resulted in the expression of ISGs (OAS3, ISG15, IFIT1), demonstrating the functionality of the produced hIFNλ1. While IFIT1 and ISG15 mRNAs showed the same expression pattern as IFNλ1, OAS3 mRNA remained elevated for up to 48 h (Fig. 4B). Animals treated with Control-mRNA did not show changes in cytokine or ISG transcript levels (Fig. 4B). CXCL9, CXCL10, CXCL11, and IL-6 transcripts were transiently induced in the lungs of ETH47-treated mice. However, no significant increase of CXC ligands at protein level in lung or plasma was detected, suggesting no undesired activity of ETH47 at pharmacologically active doses (Fig. 4C). These data demonstrate

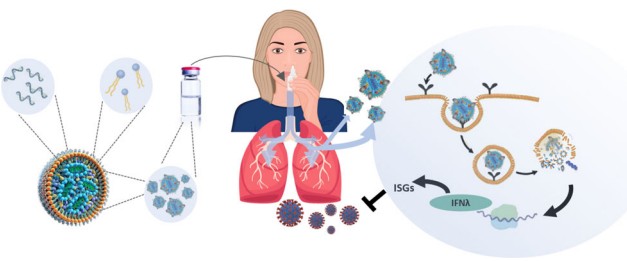

**Figure 1. Overview of ETHRIS mRNA technology.**

ETH47 (LNP-formulated IFNλ1 mRNA) drug product is intended for local delivery to the sites of virus entry and replication (nasal application via nasal spray or pulmonary application using a nebulizer).

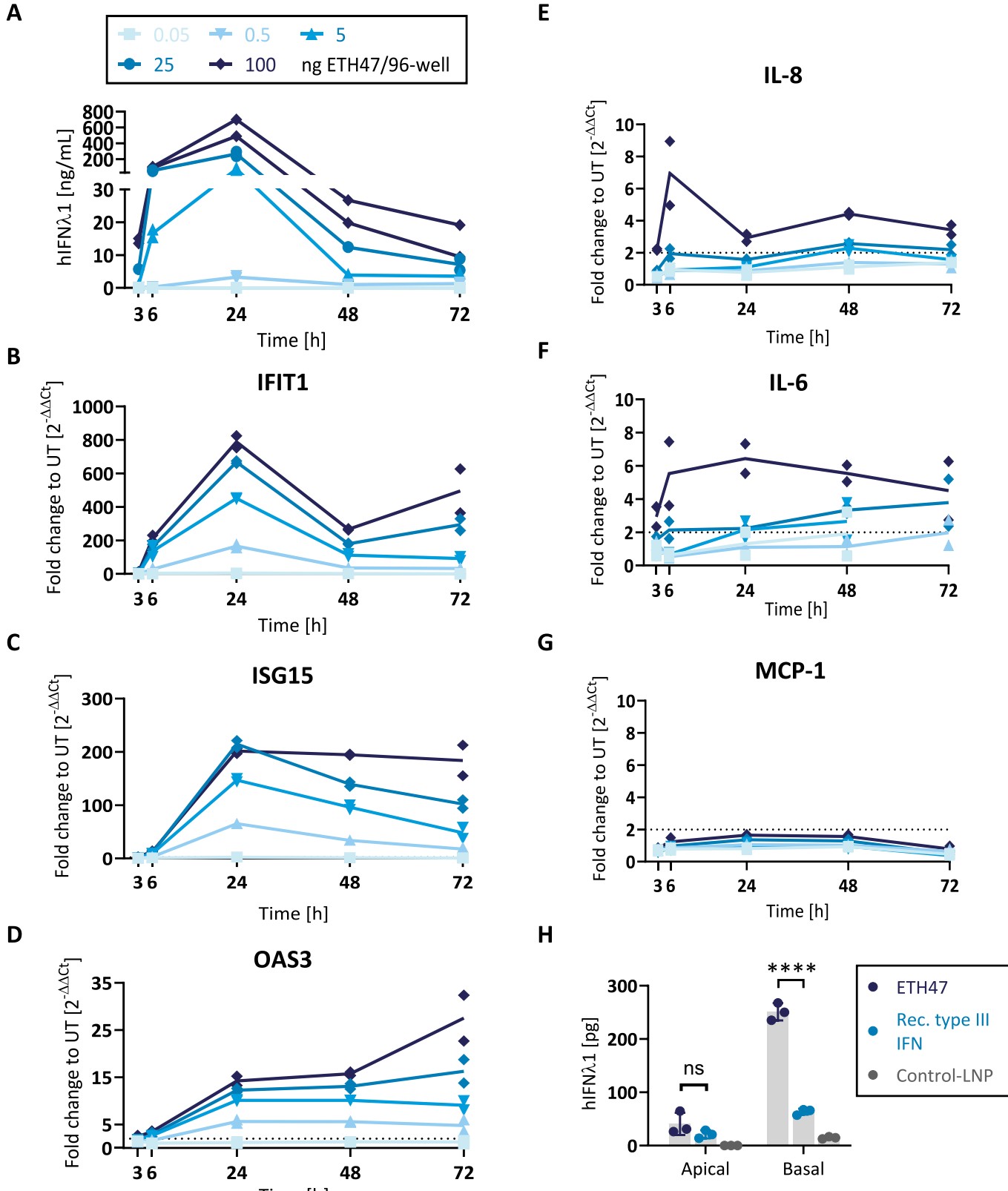

◀ **Figure 2. Single-dose treatment of A549 cells and primary human air–liquid interface cultures with ETH47 leads to dose-dependent translation of target protein and target gene induction, with little to no cytokine induction.**

(A–G) A549 cells were treated with indicated doses of ETH47 (LNP-formulated; $n = 2$ biological replicates). At 6 and 24 h post treatment, medium was exchanged. (A) hIFNλ1 protein expression in supernatants was measured via ELISA. (B–D) Target gene expression analysis via qPCR. (E–G) Cytokine expression analysis via qPCR. (H) Air–liquid interface cultures ($n = 3$ biological replicates) were treated with ETH47 (LNP-formulated mRNA) or Control-LNP (3 μg/insert each) or recombinant protein (100 ng/mL), both of which were removed at 6 h post treatment. hIFNλ1 levels were quantified 24 h post treatment in apical and basal compartments. Data Information: in (A–G) data are shown as single data points and mean. (A) All values for the lowest dose, 3 h values for the second lowest dose were BLQ, some values for highest dose were ALQ (see Source Data). (B–G) Missing data points reflect Cq values above the set cycle threshold. Dotted line at $y = 2$: fold changes below 2 are not considered as gene induction. (H) Single data points and mean $+/-$ SD are shown. ns: $P = 0.2038$; ****$P = 0.000046$ (unpaired $t$ test). Source data are available online for this figure.

desired target engagement following the administration of ETH47 in vivo and translatability of results obtained in vitro.

## ETH47 significantly reduces virus replication, body weight loss, and mortality in a hACE2-TG mouse model challenged with SARS-CoV-2

Antiviral efficacy of ETH47 was tested in transgenic K18-hACE2 mice challenged with a lethal dose of SARS-CoV-2 virus (B.1.1.7). Mice were treated with 7.5 μg/animal ETH47 or vehicle through intranasal administration 1 day prior to and 1 day following virus inoculation. Virus load was measured on day 3, and animals were followed-up for mortality until day 13. At day 3, animals treated with ETH47 showed a 27-fold ($P = 0.0031$) reduction of infectious virus particles in the lungs compared to the vehicle group (Fig. 5A). Moreover, ETH47-treated animals showed significantly lower body weight loss on days 5, 6, and 7 (Fig. 5B) as well as a significantly improved survival at day 13 compared to the vehicle group (Fig. 5C).

We next tested prophylactic antiviral efficacy by administering single-dose ETH47 one day prior to virus inoculation and observed a significant reduction of pulmonary virus load (Fig. 5D). The reduction in virus titers was corroborated by reduced body weight loss (Fig. 5E) and improved survival rates in ETH47-treated mice (Fig. 5F). Overall, this demonstrates that ETH47 has prominent antiviral and life-saving properties in a lethal SARS-CoV-2 infection model and may be used as prophylactic treatment.

Our studies show for the first time that administration of mRNA encoding IFNλ1 prior to or as a combination prior to and after virus inoculation effectively inhibits SARS-CoV-2 replication in vitro and in vivo, thereby supporting the development of ETH47 for prophylaxis of SARS-CoV-2 infection. Based on its unique mechanism of action related to modulation of the host's innate immune system, it is conceivable that ETH47 may also be effective for prophylaxis of a broader spectrum of respiratory viral infections (e.g., emerging SARS-CoV-2 or beta-coronavirus variants, influenza, respiratory syncytial virus, and others), for prophylaxis of virus-driven exacerbations in people with chronic respiratory disease, and for pandemic preparedness in the event of future emerging pandemics caused by yet unknown respiratory viruses, bridging the gap until effective vaccines become available. Further studies are needed to investigate the use of ETH47 as post exposure treatment, as well as for the treatment of respiratory viral infections.

As compared to treatment with recombinant hIFNλ1 protein, mRNA delivery resulted in superior antiviral effects in air–liquid interface cultures. This advantage of the mRNA modality may in

part be explained by the longer persistence of ETH47 as compared to recombinant protein in the lungs and higher bioavailability of IFNλ1 at the basolateral side of the epithelium. The latter observation may be particularly relevant for the therapeutic activity of ETH47 since IFN receptors are found at the basolateral site of epithelial cells (Eichinger et al, 2017; Uze et al, 2007). A further advantage of mRNA therapy is that endogenously produced IFN, in contrast to exogenously generated recombinant protein, is unlikely to elicit anti-drug antibodies that may impact upon PK, safety or efficacy (Martin et al, 2002). Our data indicate that ETH47 and Control-mRNA did not result in broad systemic immune activation, suggesting that both chemically modified mRNA as well as LNP formulation are not systemically bioavailable and immunologically inert at the tested doses. Nevertheless, key aspects revolving around the immune response remain to be investigated. Among these, peripheral tolerance of ETH47 due to sensibilization by mRNA vaccines should be addressed. Equally important, it remains to be tested whether multiple administrations could trigger B- and T-cell response against IFNλ1, as a collateral result of human protein overexpression in rodents. Furthermore, additional effects of ETH47-derived IFNλ1 other than direct ISG induction should be explored, e.g., the induction of Th1 polarization, known to be mediated by type III interferons. While further biodistribution and immunological studies are required to fully elucidate PK properties, observations made in this study generally support the concept of targeted administration of formulated modified mRNA into the lung to avoid potential undesired systemic effects.

Although the small sample size of some experiments could be a potential limitation of this study, we believe that the general reproducibility of results across experiments supports our mechanistic conclusions following ETH47 treatment. Moreover, results were further validated in the virus challenge study in mice, where we relied on a more robust sample size to prove the beneficial activity of ETH47.

Collectively, our data demonstrate the antiviral efficiency of formulated hIFNλ1 mRNA against SARS-CoV-2 and provides a rational to further develop mRNA-based immunomodulatory therapies.

## Methods

### mRNA and LNP formulation

ETH47 represents an IFNλ1 coding mRNA which was chemically modified (Kormann et al, 2011) and LNP-formulated (Jarzebinska et al, 2016).

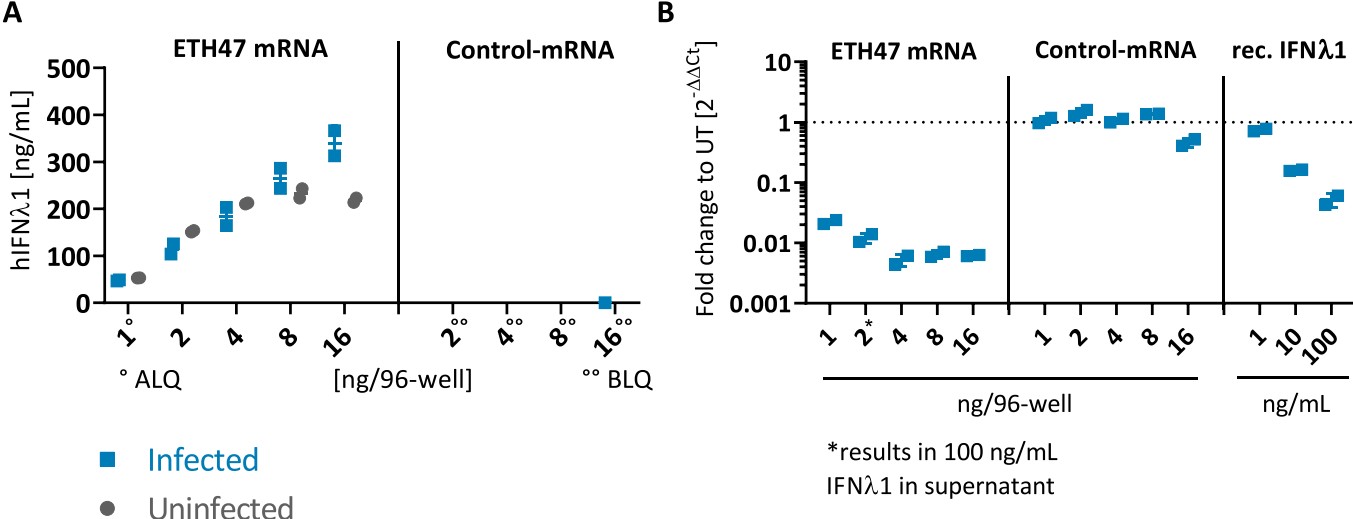

**Figure 3. ETH47 results in potent inhibition of SARS-CoV-2 virus replication in vitro.**

Lipofectamine transfection of unformulated ETH47 mRNA in A549-ACE2 cells was done 24 h prior to infection with SARS-CoV-2 (MOI 3). Harvest was done at 48 h post infection ($n = 2$ biological replicates). (**A**) hIFNλ1 ELISA from supernatants. ALQ above upper limit of quantification (sample dilution resulted in a value above ULOQ, real concentration might be slightly different), BLQ below lower limit of quantification. (**B**) qPCR for virus load. Dotted line at $y = 1$ is untreated reference. Asterisk indicates transfection dose that results in approximately 100 ng/mL produced protein in the supernatant. Data Information: data are presented as individual data points and mean $+/-$ SD. Source data are available online for this figure.

ETH47 mRNA is a chemically modified mRNA encoding hIFNλ1 with an average length of 788 nucleotides. The mRNA is manufactured by in vitro transcription (IVT) from a linearized plasmid deoxyribonucleic acid (DNA) template using a T7 RNA polymerase. The chemically modified bases (5-methylcytosine and 2-thiouridine) are introduced during IVT to generate partially modified mRNA. The complete structure of the mRNA includes a 5' Cap, a 5' minimal UTR, the codon-optimized coding sequence, and a poly(A) tail.

ETH47 is a preservative-free, sterile dispersion of that mRNA that is formulated in lipidoid nanoparticles (LNP) in an aqueous cryoprotectant solution. The lipidoid nanoparticles are generated by mixing the mRNA with a solution of one lipidoid and three lipid excipients. The excipients associate with the mRNA, protect it from degradation, and aid its delivery to the target cells in the respiratory epithelium. Downstream processing after mixing lipids and mRNA includes buffer exchange, concentration and sterile filtration.

## Cells

A549 and A549-ACE2 cells were used for in vitro experiments. A549 cells were purchased from DSMZ (ACC107, human lung epithelial cells) A549-ACE2 cells were described previously (Stukalov et al, 2021). Authentication is done by DNA profiling using STR for A549 and A549-ACE2.

## In vitro transfection

Transfection with mRNA was done using Lipofectamine® MessengerMAX™ (Thermo Fisher Scientific, LMRNA008) in a mRNA to Lipofectamine ratio of 1:1.5 (w/v). Formulated mRNA was diluted with vehicle solution (10% (w/v), 50 mM NaCl and a proprietary

excipient) to the highest dose and dose titration was performed in medium. In total, 25 μL of desired doses were added to the respective wells after medium exchange.

## Air–liquid interface (ALI) cultivation

MucilAir™ inserts were purchased at Epithelix SàRL. No STR profiling was done. Cells were cultivated in 700 μL MucilAir™ culture medium at 37 °C, in a humidified atmosphere with 5% $CO_2$ as air–liquid interface (ALI) cultures. Prior transfection the cells were let rest for 2–3 days upon arrival. ALI cultures were transfected with 3 μg ETH47 per insert. Basal medium and apical wash were quantified for hIFNλ1. To maintain culture, medium was changed every 2–3 days.

## Viruses

Virus strains of SARS-CoV-2 used in this study are publicly accessible through GenBank: https://www.ncbi.nlm.nih.gov/genbank/ platform under the accession number MZ314997 (B.1.1.7) and MT270101 (Wuhan).

For in vitro virus infection, we used SARS-CoV-2 isolate from the following study: https://www.ebi.ac.uk/ena/browser/view/PRJEB38744.

## Virus infection in vitro

SARS-CoV-2-MUC-IMB-1 was produced by infecting VeroE6 cells cultured in DMEM medium (10% FCS, 100 μg/ml Streptomycin, 100 IU/ml Penicillin) for 2 days (MOI 0.01). Viral stock was harvested and spun twice (1000 g/10 min) before storage at −80 °C. The titer of viral stock was determined by plaque assay. For this,

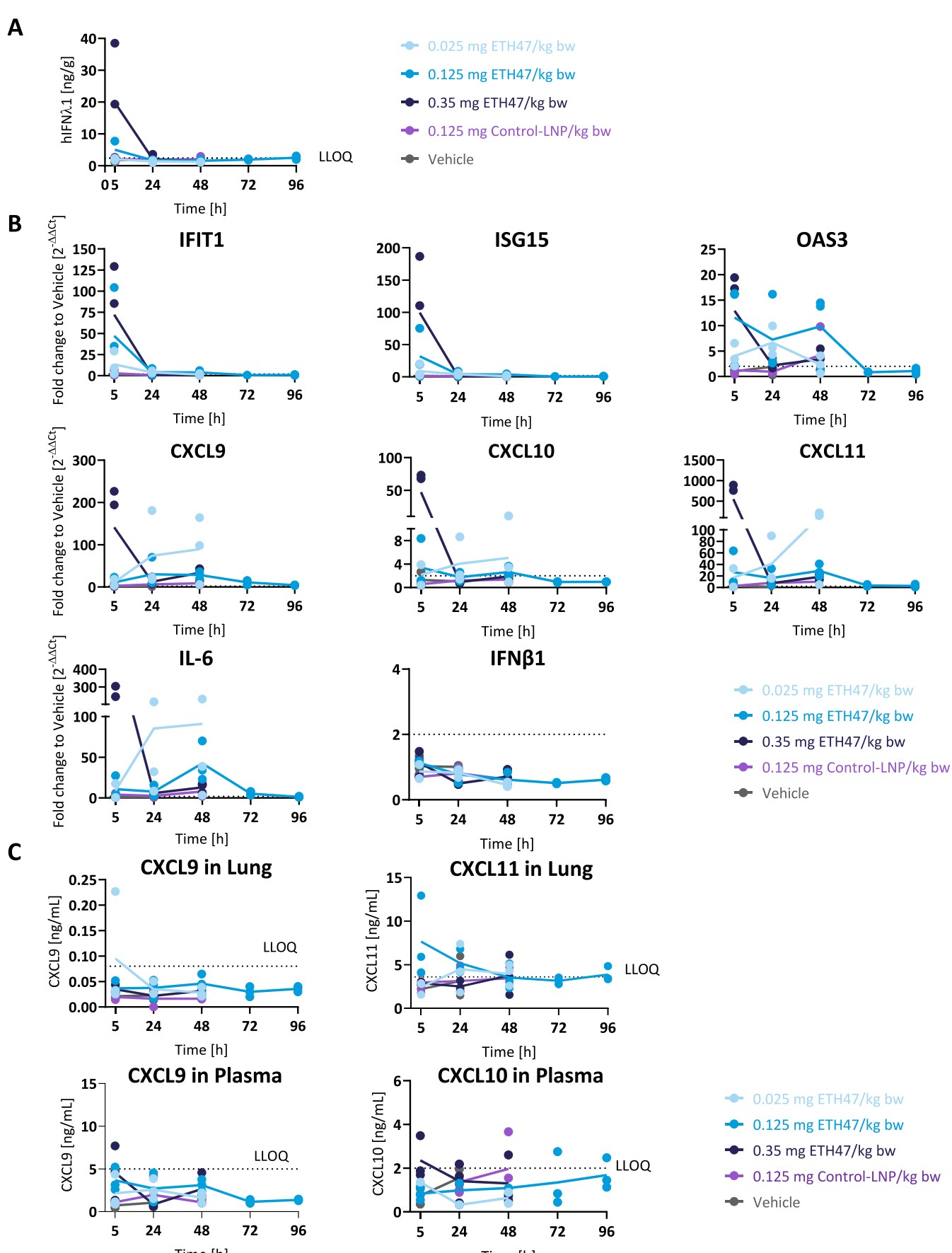

◄ **Figure 4.  Single nasal administration of ETH47 induces pulmonary protein and target gene expression in mice.**

Necropsy was done at 5, 24, and 48 h for all groups. In addition, 0.35 mg ETH47/kg bw group also included 72 and 96 h timepoints. Animals in vehicle group were sacrificed at 5 or 24 h post application. $n = 3$ biological replicates for each dose and timepoint. (A) hIFNλ1 protein expression in lung homogenates. One sample of 0.125 mg ETH47/kg bw group was excluded due to insufficient sample quality. (B) qPCR for target genes in lung homogenates. (C) Chemokine ELISA in plasma and lung homogenate. LLOQ lower limit of quantification, Bw body weight. Data Information: individual data points and mean are shown. Source data are available online for this figure.

confluent monolayers of VeroE6 cells were infected with serial fivefold dilutions of virus supernatants for 1 h at 37 °C. The inoculum was removed and replaced with serum-free MEM containing 0.5% carboxymethylcellulose. Two days post infection, cells were fixed for 20 min at room temperature with formaldehyde directly added to the medium to a final concentration of 5%. Fixed cells were washed extensively with PBS before staining with $H_2O$ containing 1% crystal violet and 10% ethanol for 20 min. After rinsing with PBS, the number of plaques was counted, and the virus titer was calculated. Cells were infected at indicated timepoints with MOI 3. Virus was added in 25 μL medium directly to the cell culture.

## RT-qPCR

For RNA isolation, cells were lysed using the SingleShot™ Cell Lysis Kit according to the manufacturer's protocol. For cDNA synthesis iScript™ Select cDNA Synthesis Kit (Bio-Rad, 1708897) was used according to manufacturer's instructions using Oligo(dT) primers. For qPCR nuclease free water, TaqMan probe (Thermo Fisher Scientific) and TaqMan fast advanced Master Mix (Thermo Fisher Scientific) were mixed. 18 μL of TaqMan Master Mix were transferred to an optical 96-well qPCR reaction plate. 2 μL cDNA template (cDNA and nuclease free water) were added to obtain a final volume of 20 μL. Analysis of qPCR results was done with the ΔΔCt method. All used TaqMan Probes are shown in Table 1.

## qPCR for SARS-CoV-2 mRNA

For relative transcript quantification, PowerUp SYBR Green was used. All steps were performed according to the manufacturer's instructions. RPLP0 was employed as a reference gene. For primer sequences, refer to Table 2.

## IFNλ1 ELISA

ELISA was performed according to kit protocol for human IL29 ELISA (Abcam, ab100568) with the exception that TMB incubation time was shortened to 10 min. Absorbance is measured at 450 nm with reference wavelength of 650 nm. The analysis is done upon interpolation of delta OD using a 4PL standard curve. GraphPad Prism V.8 is used for analysis.

## CXCL9 and CXCL10 ELISA in plasma

Plasma samples were measured according to the manufacturer's instructions for CXCL9 (Mouse CXCL9/MIG DuoSet ELISA; R&D Systems; DY492-05) and CXCL10 (Mouse CXCL10/IP-10/CRG-2 DuoSet ELISA; R&D Systems; DY466). Deviations to the protocol are shown in Table 3.

## CXCL9 and CXCL11 ELISA in lung homogenate

Lung homogenates were measured according to the manufacturer's instructions for CXCL9 (Mouse CXCL9/MIG DuoSet ELISA; R&D Systems; DY492-05) and CXCL11 (Mouse CXCL11/I-TAC DuoSet ELISA; R&D Systems; DX572). Deviations to the protocol are shown in Table 4.

## Animal housing

All experiments with C57/BL6 were approved in advance by the local animal welfare authorities (Regierung von Oberbayern) under the file number Az. 2532.Vet_03-17-114 and were conducted according to the German animal protection law (Tierschutzgesetz). Mice were housed under specific pathogen-free conditions (facility tested negative for any FELASA-listed pathogens according to the annual health and hygiene survey 2017) in individually ventilated cages under a circadian light cycle (lights on from 7 a.m. to 7 p.m.). Female C57/BL6 were obtained from Janvier, France. Food and drinking water were provided ad libitum. After arrival, animals were given 7 days for acclimatization until they entered the study.

All experiments with K18-hACE c57BL/6J mice were approved in advance by the Animal Ethics Committee at the Danish Veterinary and Food Administration (Stationsparken 31-33, 2600 Glostrup, Denmark) and were carried out in accordance with the Danish Animal Welfare Act for the Care and Use of Animals for Scientific Purposes. In all, 6–8 weeks old female K18-hACE c57BL/6J mice (strain: 2B6.Cg-Tg(K18-ACE2)2Prlmn/J) were obtained from The Jackson Laboratory. Mice were fed with standard chow diet and were housed in pathogen-free facility.

## Administration of test item to C57/BL6 mice

Animals were anesthetized by inhalation of pure oxygen at a flow rate of 2 L/min supplemented with approximately 4% of the anesthetic gas Isoflurane in an inhalation chamber. Test item was applied in two 25 μL boluses, by using a laboratory pipette, on both nose holes so that the liquid was actively inhaled through the nose by the animal during a physiological inspiratory movement. The animal was held in perpendicular position during this procedure and was subsequently placed in supine position until recovery from anesthesia.

## Administration of test item and SARS-CoV-2 virus

Animals were sedated by inhalation of air supplemented with isoflurane gas (~2–4%). 15 μL of test item was applied in two fractions of 7.5 μL on each nostril of the animal and inhaled during physiological inspiratory movement. Immediately after administration, animals were placed in their home cages for recovery. For

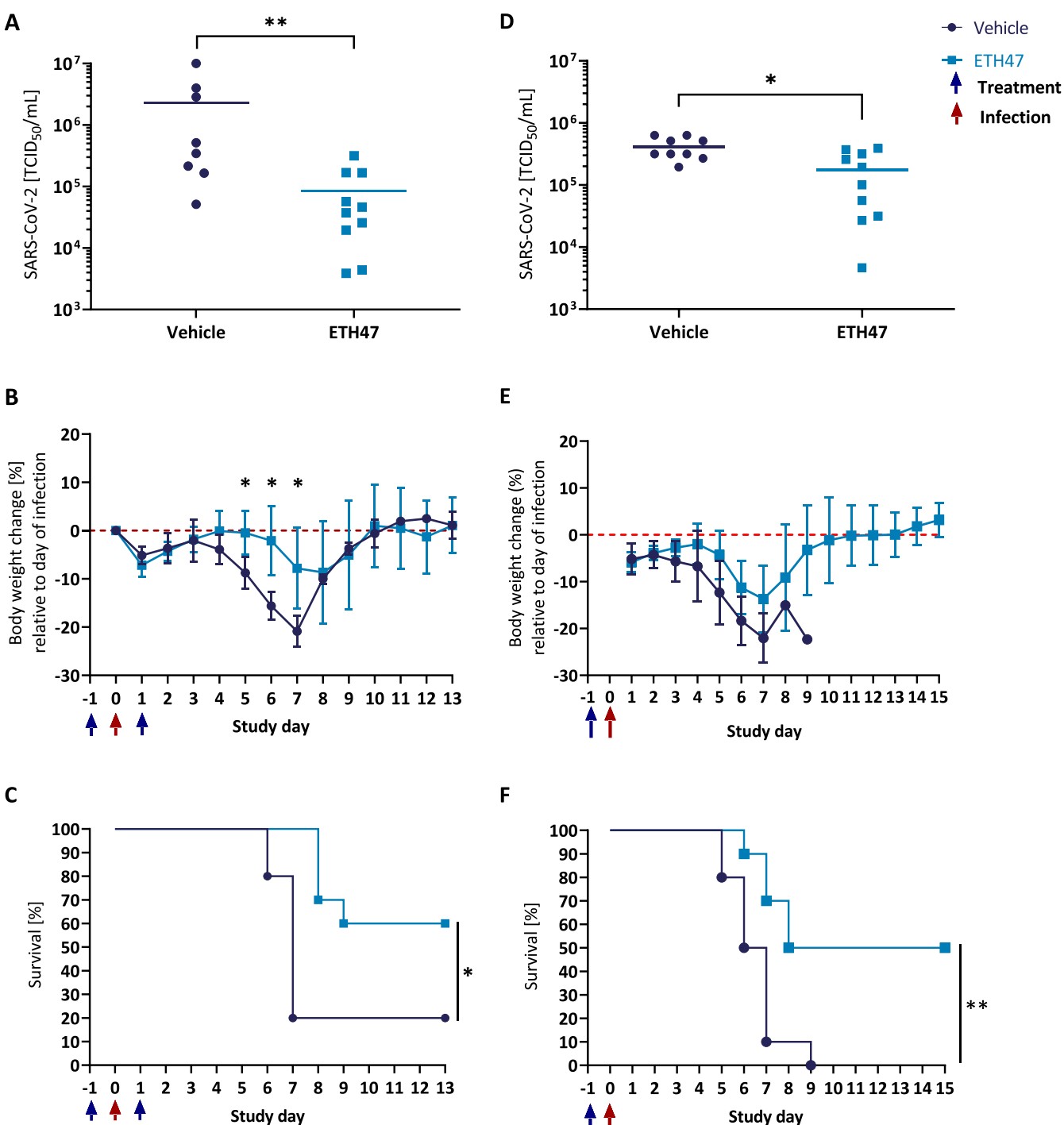

Figure 5.   ETH47 reduces virus replication, body weight reduction, and mortality in a hACE2-Transgenic mouse model challenged with SARS-CoV-2.

(A–C) Intranasal administration of 0.38 mg ETH47/kg bw 1 day before and 1 day after SARS-CoV-2 infection (2500 PFU). (D–F) Intranasal administration of 0.38 mg ETH47/kg bw one day before SARS-CoV-2 infection. (A, D) $TCID_{50}$ assay in lung homogenate ($n = 10$ individuals). Note that two animals of the vehicle group of panel (A) were excluded from the analysis due to humane reasons (being unable to eat due to elongated teeth) or processing issues during organ sampling, which led to high amounts of residual blood in the lung and precluded a proper downstream analysis. (B, E) Percentual weight change ($n = 20$ biological replicates until day 3, $n = 10$ from day 4 on). (C) Kaplan–Meier estimation ($n = 20$ biological replicates until day 3, $n = 10$ from day 4 on). (F) Kaplan–Meier estimation ($n = 10$ biological replicates). Bw body weight. Data Information: (A, D) Individual data points and mean are shown. *$P = 0.0122$, **$P = 0.0031$ (Mann–Whitney U test). (B, E) Data are shown as mean $+/-$ SD. *$P < 0.05$; only significant comparisons are indicated, all other comparisons are nonsignificant (Mann–Whitney U test). (C, F) *$P = 0.0113$, **$P = 0.003$ (curve-comparison with Log-rank (Mantel–Cox) test). Source data are available online for this figure.

SARS-CoV-2 virus infection, 2500 PFU/mouse were inoculated under anesthesia (75 μg/g Ketamin, and 1 μg/g Medetomidine), and the anesthesia was partially antagonized by administration of 1 μg/g Atipamezol.

## SARS-CoV-2 propagation

A clinical isolate of B.1.1.7 SARS-CoV2 (Kent, UK) was provided under MTA by Professor Arvind Patel, University of Glasgow. The B.1.1.7 variant is listed in GenBank under accession number MZ314997. The virus was propagated in VeroE6 cells expressing human TMPRSS2 (VeroE6-hTMPRSS2) (kindly provided by Professor Stefan Pöhlmann, University of Göttingen) (Hoffmann et al, 2020). Briefly, VeroE6-hTMPRSS2 cells were infected with a multiplicity of infection (MOI) of 0.05, in DMEM (Gibco) + 2% FCS (Sigma-Aldrich) + 1% Pen/Strep (Gibco) + L-Glutamine (Sigma-Aldrich) (from here, complete medium). 72 h post infection, supernatant (containing new virus progeny) was harvested and concentrated on 100 kDa Amicon ultrafiltration columns (Merck) via centrifugation at 4000 G for 30 min. Virus titer was determined using $TCID_{50}$% assay and was calculated using the Reed-Muench method (Muench, 1938). To convert to the mean number of plaque forming units (pfu)/mL, the $TCID_{50}$/mL was multiplied by the factor 0.7 (ATCC— Converting TCID [50] to plaque forming units (PFU)).

## Clinical examination and survival analysis

Body weight development (in relation to study day 0) was the main read-out for determining the animals' clinical condition and was performed during the entire time course of the study. Consequently, animals were weighed daily beginning from the start of the study (i.e., first treatment using test item or/and infection with the virus). Percentual weight loss of more than 20% or clinical signs of more than moderate suffering were considered as humane endpoint. Resulting survival times were analyzed using the Kaplan–Meier estimation and were plotted in form of a Kaplan–Meier curve.

## $TCID_{50}$ assay

Lungs were cut in small pieces and homogenized with steel beads in a TissueLyser (II) (both from QIAGEN) in 1 ml PBS at 30 Hz for 6 min and centrifuged $300 \times g$ for 1 min to avoid debris. In total, 10 μl of the supernatant were used for $TCID_{50}$ assay.

To determine the amount of infectious virus in cell culture supernatant or generated virus stocks, a limiting dilution assay was performed. In all, $2 \times 10^4$ VeroE6-TMPRRS2 cells were seeded in 90 μL DMEM in a 96-well plate. The following day, samples were

**Table 1. TaqMan probes.**

| Target | Supplier | Cat no. |
|---|---|---|
| **Human** | | |
| IFIT1 | Thermo Fisher Scientific | Hs03027069_s1 |
| OAS3 | Thermo Fisher Scientific | Hs00196324_m1 |
| ISG15 | Thermo Fisher Scientific | Hs01921425_s1 |
| Mx1 | Thermo Fisher Scientific | Hs008955608_m1 |
| IL-8 | Thermo Fisher Scientific | Hs00174103_m1 |
| IL- 6 | Thermo Fisher Scientific | Hs00174131_m1 |
| RPLP0 | Thermo Fisher Scientific | Hs00420895_gH |
| ACTB | Thermo Fisher Scientific | Hs01060665_g1 |
| MCP-1 | Thermo Fisher Scientific | Hs00234140_m1 |
| **Murine** | | |
| CXCL9 | Thermo Fisher Scientific | Mm00434946_m1 |
| CXCL10 | Thermo Fisher Scientific | Mm00445235 |
| CXCL11 | Thermo Fisher Scientific | Mm00444662_m1 |
| IFNb1 | Thermo Fisher Scientific | Mm00439552_s1 |
| IFIT1 | Thermo Fisher Scientific | Mm07295796_m1 |
| ISG15 | Thermo Fisher Scientific | Mm01705339_s1 |
| OAS3 | Thermo Fisher Scientific | Mm00460944_m1 |
| IL-6 | Thermo Fisher Scientific | Mm00446190_m1 |
| RPLP0 | Thermo Fisher Scientific | Mm00725448_s1 |

**Table 2. Primer Sequences for virus qPCR.**

| Target | Primer | Forward | Reverse |
|---|---|---|---|
| CoV2 | 2019-nCoV_N2 | TTACAAACATTGGCCGCAAA | GCGCGACATTCCGAAGAA |
| RPLP0 | RPLP0 | GGATCTGCTGCATCTGCTTG | GCGACCTGGAAGTCCAACTA |

**Table 3. Deviations from the manufacturer's instructions for CXCL ELISAs in plasma.**

| Cytokine | Blocking solution | Antibody and sample diluent | TMB incubation time | Sample dilution | LLOQ |
|---|---|---|---|---|---|
| CXCL9 | Casein | 0.05% Tween-20 in 1× PBS | 16 min | 1:2 | 40 pg/mL |
| CXCL10 | 1% BSA in 1x PBS | 1% BSA in 1× PBS | 25 min | 1:20 | 100 pg/mL |

**Table 4. Deviations from manufacturer's instructions for CXCL ELISAs in lung homogenate.**

| Cytokine | Blocking solution | Antibody and sample diluent | TMB incubation time | Sample dilution | LLOQ |
|---|---|---|---|---|---|
| CXCL9 | Casein | 0.05% Tween-20 in 1× PBS | 16 min | 1:50 | 100 pg/mL |
| CXCL11 | Casein | 0.05% Tween-20 in 1× PBS | 16 min | 1:30 | 120 pg/mL |

thawed and diluted tenfold. Subsequently, samples were serially diluted tenfold using DMEM. Cells were incubated for 72 h at 37 °C and 5% $CO_2$. Cytopathic effect (CPE) was scored after cell fixation using 5% formalin (Sigma-Aldrich) and crystal violet staining (Sigma-Aldrich), using a light microscope (Leica DMi1). The tissue culture infectious dose 50 ($TCID_{50}$/mL) was finally calculated using the method of Reed and Muench.

## Statistics and reproducibility

Sample sizes of $n = 2$ and $n = 3$ were used for in vitro studies. At least $n = 3$ was used for in vivo studies. In vivo challenge studies were done with at least $n = 10$. Experimental procedures were well-established before. Sufficient material was available for analysis of each sample. Statistical differences between the groups were calculated using Mann–Whitney's nonparametric $U$ test or unpaired $t$ test. A $P$ value smaller than 0.05 was considered as a statistically significant difference. A survival analysis was performed using the Kaplan–Meier estimation. As for the $U$ test, a $P$ value smaller than 0.05 was considered as statistically significant difference between ETH47 and vehicle-treated groups. All types of analysis were performed using GraphPad Prism software.

## Biosafety

All in vitro experiments with SARS-CoV-2 were performed at the Institute of Virology, Technical University of Munich, School of Medicine, Munich, Germany) in BSL3 laboratories under the approval of the "Regierung von Oberbayern", Germany (AZ: 55.1GT-8791.GT_2-365-10 and 55.1GT-8791.GT_2-365-20).

All in vivo experiments with SARS-CoV-2 were performed at the Department Biomedicine, Aarhus University in a biosafety level 2+ laboratory. All aspects of this study are approved by the office of Danish Working Environment Authority, Landskronagade 33, 2100 Copenhagen Ø, before initiation of this study.

# Data availability

No primary datasets have been generated or deposited.

The source data of this paper are collected in the following database record: biostudies:S-SCDT-10_1038-S44319-024-00216-4.

# Peer review information

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

## Acknowledgements

The authors thank Ea S Andersen for technical assistance with the work related to the animal infection experiments. The authors thank N. Tomkinson and E. Murenu for valuable feedback on the manuscript. AM, KL, EAP, EB, AS, WZ, AW, ELH, SH, GH, SM, CR and TL were funded by "Bayern Innovativ" (grant number TPP-2103-0005).

## Author contributions

**Anna Macht**: Formal analysis; Investigation; Visualization; Writing—original draft; Writing—review and editing. **Yiqi Huang**: Investigation. **Line S Reinert**: Investigation. **Vincent Grass**: Investigation. **Kristin Lohmer**: Investigation. **Elke Tatjana Aristizabal Prada**: Investigation. **Eveline Babel**: Investigation. **Alexandra Semmler**: Investigation. **Wen Zhang**: Investigation. **Andrea Wegner**: Investigation. **Eva Lichtenegger-Hartl**: Project administration. **Sonja Haas**: Supervision; Writing—original draft; Writing—review and editing. **Günther Hasenpusch**: Formal analysis; Investigation. **Steffen Meyer**: Project administration. **Søren R Paludan**: Supervision; Writing—review and editing. **Andreas Pichlmair**: Supervision; Writing—review and editing. **Carsten Rudolph**: Conceptualization; Supervision; Project administration; Writing—review and editing. **Thomas Langenickel**: Conceptualization; Supervision; Project administration; Writing—review and editing.

Source data underlying figure panels in this paper may have individual authorship assigned. Where available, figure panel/source data authorship is listed in the following database record: biostudies:S-SCDT-10_1038-S44319-024-00216-4.

## Disclosure and competing interests statement

AM, KL, EAP, EB, AS, WZ, AW, ELH, SH, GH, SM, and TL are employees of ETHRIS GmbH. CR is shareholder of ETHRIS GmbH.

# Expanded View Figure

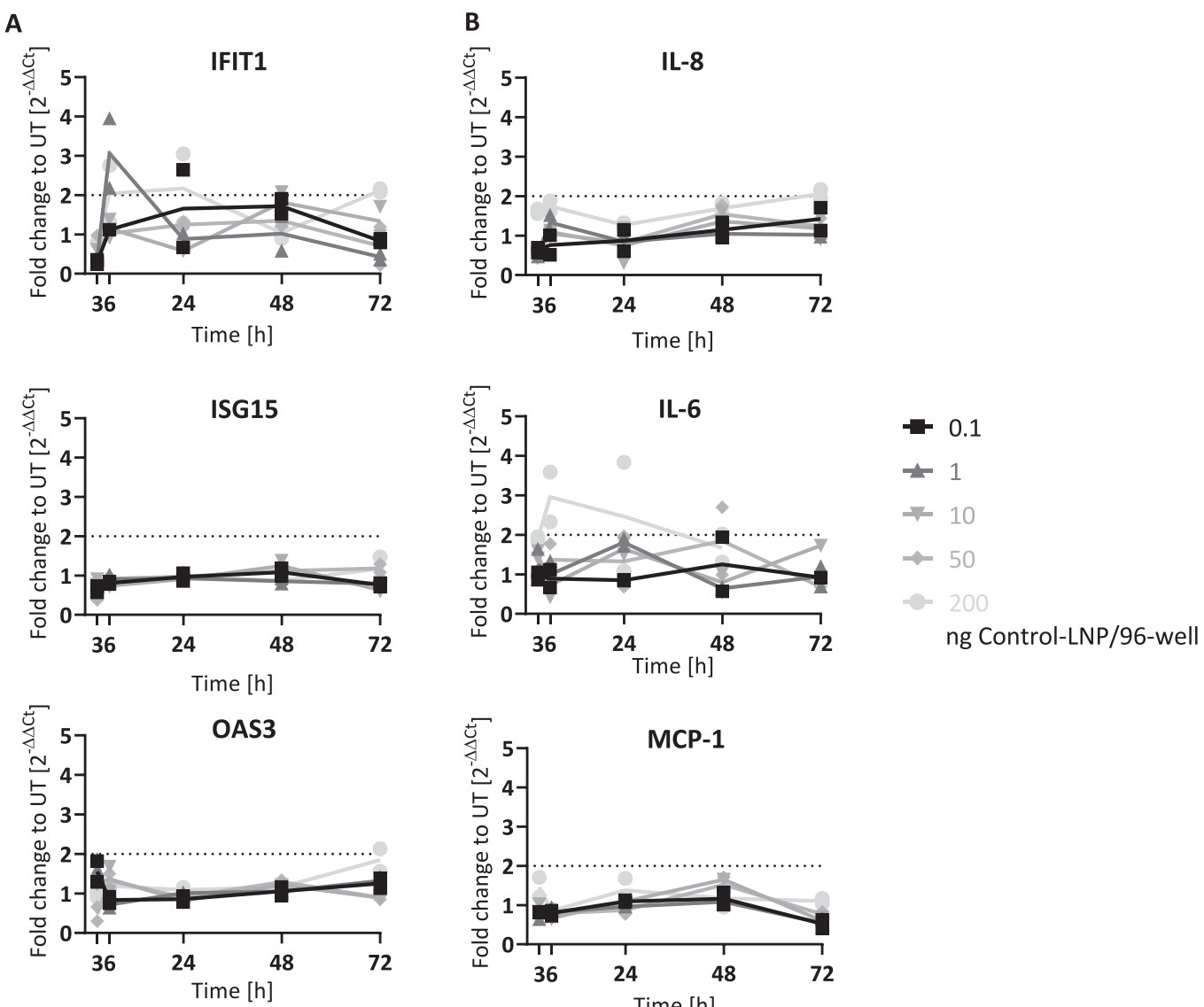

**Figure EV1. Single-dose treatment of A549 cells with Control-LNP leads to no or very low target gene induction and no cytokine induction.**

A549 cells were treated with Control-LNP ($n = 2$ biological replicates). At 6 and 24 h post treatment medium was exchanged (**A**) Target gene and (**B**) cytokine expression in A549 cells. Data information: data points show single values and mean. Dotted line at $y = 2$: fold changes below 2 are not considered as gene induction. Missing data points reflect Cq values above the set cycle threshold. Source data are available online for this figure.

