## [Peer Review File · EMBO Reports]

Mucosal IFN λ 1 mRNA-based immunomodulation effectively reduces SARS-CoV-2 induced mortality in mice

Anna Macht, Yiqi Huang, Line Reinert, Vincent Grass, Kristin Lohmer, Elke Aristizabal-Prada, Eveline Babel, Alexandra Semmler, Wen Zhang, Andrea Wegner, Eva Lichtenegger-Hartl, Sonja Haas, Günther Hasenpusch, Steffen Meyer, Søren Paludan, Andreas Pichlmair, Carsten Rudolph, and Thomas Langenickel

Corresponding author(s): Thomas Langenickel (langenickel@ethris.com)

Review Timeline:

Submission Date:	16th Jan 24
Editorial Decision:	8th Feb 24
Revision Received:	9th May 24
Editorial Decision:	23rd May 24
Revision Received:	12th Jul 24
Accepted:	15th Jul 24

Editor: Achim Breiling

Transaction Report:

Dear Dr. Langenickel,

Thank you for the submission of your research manuscript to EMBO reports. I have now received the reports from the three referees that were asked to evaluate your study, which can be found at the end of this email.

As you will see, the referees think that the findings are of interest. However, they have several comments, concerns, and suggestions, indicating that a major revision of the manuscript is necessary to allow publication of the study in EMBO reports. As the reports are below, and all the referee concerns need to be addressed, I will not detail them here. Referee #1 indicates that a more virology-oriented journal might be better suited for publication of the study. However, after looking at the positive evaluations of the other referees and after further editorial assessment in the team, we consider the general interest in these data as high and decided to proceed with the manuscript at EMBO reports.

Given the constructive referee comments, I would like to invite you to revise your manuscript with the understanding that all referee concerns must be addressed in the revised manuscript or in a detailed point-by-point response. Acceptance of your manuscript will depend on a positive outcome of a second round of review. It is EMBO reports policy to allow a single round of revision only and acceptance of the manuscript will therefore depend on the completeness of your responses included in the next, final version of the manuscript.

Of note, please provide more information (in particular in the methods section), on the design and synthesis of ETH47. Moreover, please also add a paragraph titled 'Biosafety' to the methods section gathering all information on where and how biosafety-relevant experiments with viruses were performed and that these were approved, and by whom (institution, government).

- 1) a .docx formatted version of the final manuscript text (including legends for main figures, EV figures and tables), but without the figures included. Figure legends should be compiled at the end of the manuscript text.
- 2) individual production quality figure files as .eps, .tif, .jpg (one file per figure), of main figures (up to 5 for a report - see below) and EV figures. Please upload these as separate, individual files upon re-submission.

For more details, please refer to our guide to authors:
<http://www.embopress.org/page/journal/14693178/authorguide#manuscriptpreparation>

Please consult our guide for figure preparation:
http://wol-prod-cdn.literatumonline.com/pb-assets/embo-site/EMBOPress_Figure_Guidelines_061115-1561436025777.pdf

See also the guidelines for figure legend preparation:
<https://www.embopress.org/page/journal/14693178/authorguide#figureformat>

3) We would publish your manuscript in the Report format (as also indicated by you in the submission system). For a Scientific Report we require that results and discussion sections are combined in a single chapter called "Results and Discussion". Please do this for your manuscript. For more details, please refer to our guide to authors:
<http://www.embopress.org/page/journal/14693178/authorguide#researcharticleguide>

5) a complete author checklist, which you can download from our author guidelines (<https://www.embopress.org/page/journal/14693178/authorguide>). Please insert page numbers in the checklist to indicate where the requested information can be found in the manuscript. The completed author checklist will also be part of the RPF.

Please also follow our guidelines for the use of living organisms, and the respective reporting guidelines: <http://www.embopress.org/page/journal/14693178/authorguide#livingorganisms>

6) that primary datasets produced in this study (e.g. RNA-seq, ChIP-seq, structural and array data) are deposited in an appropriate public database. If no primary datasets have been deposited, please also state this in a dedicated section (e.g. 'No primary datasets have been generated and deposited'), see below.

The accession numbers and database should be listed in a formal "Data Availability" section (placed after Materials & Methods) that follows the model below. This is now mandatory (like the COI statement). Please note that the Data Availability Section is restricted to new primary data that are part of this study. This section is mandatory. As indicated above, if no primary datasets have been deposited, please state this in this section

Data availability

7) We now request the publication of original source data with the aim of making primary data more accessible and transparent to the reader. Our source data coordinator will contact you to discuss which figure panels we would need source data for and will also provide you with helpful tips on how to upload and organize the files.

8) Our journal encourages inclusion of *data citations in the reference list* to directly cite datasets that were re-used and obtained from public databases. Data citations in the article text are distinct from normal bibliographical citations and should directly link to the database records from which the data can be accessed. In the main text, data citations are formatted as follows: "Data ref: Smith et al, 2001" or "Data ref: NCBI Sequence Read Archive PRJNA342805, 2017". In the Reference list, data citations must be labeled with "[DATASET]". A data reference must provide the database name, accession number/identifiers and a resolvable link to the landing page from which the data can be accessed at the end of the reference. Further instructions are available at: <http://www.embopress.org/page/journal/14693178/authorguide#referencesformat>

9) Regarding data quantification and statistics, please make sure that the number "n" for how many independent experiments were performed, their nature (biological versus technical replicates), the bars and error bars (e.g. SEM, SD) and the test used to calculate p-values is indicated in the respective figure legends (also for EV figures and all those in an Appendix). Please also check that all the p-values are explained in the legend, and that these fit to those shown in the figure. Please provide statistical testing where applicable. Please avoid the phrase 'independent experiment', but clearly state if these were biological or technical replicates. Please also indicate (e.g. with n.s.) if testing was performed, but the differences are not significant. In case n=2, please show the data as separate datapoints without error bars and statistics. See also: <http://www.embopress.org/page/journal/14693178/authorguide#statisticalanalysis>

10) Please add scale bars of similar style and thickness to microscopic images, using clearly visible black or white bars (depending on the background). Please place these in the lower right corner of the images themselves. Please do not write on or near the bars in the image but define the size in the respective figure legend.

11) Please also note our reference format:

12) We updated our journal's competing interests policy in January 2022 and request authors to consider both actual and perceived competing interests. Please review the policy <https://www.embopress.org/competing-interests> and update your competing interests if necessary. Please name this section 'Disclosure and Competing Interests Statement' and put it after the Acknowledgements section.

13) We now use CRediT to specify the contributions of each author in the journal submission system. CRediT replaces the author contribution section. Please use the free text box to provide more detailed descriptions and do not provide your final manuscript text file with an author contributions section. See also our guide to authors: <https://www.embopress.org/page/journal/14693178/authorguide#authorshippinguidelines>

14) We would encourage you to use 'Structured Methods', our new Materials and Methods format. According to this format, the Materials and Methods section should include a Reagents and Tools Table (listing key reagents, experimental models, software and relevant equipment and including their sources and relevant identifiers), uploaded as separate file, followed by a Methods and Protocols section in which we encourage the authors to describe their methods using a step-by-step protocol format with bullet points, to facilitate the adoption of the methodologies across labs. More information on how to adhere to this format as well as downloadable templates (.doc or .xls) for the Reagents and Tools Table can be found in our author guidelines (section 'Structured Methods'):

15) Please add up to 5 keywords to the manuscript and order the sections like this, using these names:
Title page - Abstract - Keywords - Introduction - Results and Discussion - Materials and Methods - Data availability section - Acknowledgements - Disclosure and Competing Interests Statement - References - Figure legends - Expanded View Figure legends

Finally, please note that all corresponding authors are required to supply an ORCID ID for their name upon submission of a revised manuscript. Please find instructions on how to link the ORCID ID to the account in our manuscript tracking system in our Author guidelines: <http://www.embopress.org/page/journal/14693178/authorguide#authorshippinguidelines>

I look forward to seeing a revised version of your manuscript when it is ready. Please let me know if you have questions or comments regarding the revision.

Yours sincerely,

Referee #1:

In this manuscript, Macht et al. investigated the mRNA-based antiviral immunotherapeutic for SARS-CoV-2. Although the subject matter is not entirely novel, it is established that human IFN λ 1 serves as an antiviral effector. Furthermore, the manuscript lacks any methodological or conceptual innovations. While the manuscript is generally readable, with well-executed and interpretable figures, there are several aspects that dampen my enthusiasm for it. I do not recommend publishing this manuscript in EMBO Reports; it may be better suited for publication in a more specialized journal.

Comments:

Line 5. Remove 'among'.

Line 35. Why only select IFIT1, ISG15, and OAS3 as ISGs to detect?

Line 35. The expression of ISGs was only detected by qPCR. The protein level of those ISGs should be determined.

Line 39-40. The cytokines can induce cytokine release syndrome associated with the severity of the disease. For example, tumor necrosis factor- α causes lethal lung inflammation during SARS-CoV-2 infection. Thus, the level of those cytokines should be investigated.

Line 43. Please give the reference about ALI.

Fig2 a-b. Lack of the result of Control-LNP.

Fig2 a, c and d. The statistics should be determined.

Fig2. Why the concentration of Control-LNP was different from ETH47?

Fig3 a. The figure should be polished and the statistics should be done between the ETH47 and Control-LNP at different concentrations.

Line 69. As a human protein, how much about the similarity between humans and mice support the activity of hIFN λ in mice?

Line 75-80. Why the detected mRNA and protein in animals was different from A549 cells?

Line 85-86. The concentration and times of used ETH47 in animals were different from the former concentration in Figure4, why?

Line92-96. According to Figure, the expression of most ISGs was back to basal level one day post ETH47 treatment. Administering single-dose ETH47 one day before virus inoculation could reduce virus load, how does it work?

Line 109. Do you mean that air-liquid interface cultures?

Line 153. Lack all the primers of targeted genes.

Line 162. Why does the method contain IAV?

Line 187. How old were those mice used?

Line 191. Replace 'Sars-CoV-2' with 'SARS-CoV-2'.

Line 217. What is 'DMEM5'?

Line 222. Please give the reference about 'the method of Reed and Muench'.

Line 228. In each figure legend, Mann-Whitney's non-parametric U-test or unpaired t test should be detailed.

Referee #2:

Macht et al present an original study taking advantage of the mRNA platform to deliver hIFN λ 1 as antiviral strategy. They report that the IFN λ 1 mRNA LNP (called ETH47) efficiently triggers production of the cytokine in lung epithelial cells in vitro. This IFN λ 1 produced induces downstream ISG production without over-activating inflammation. Mucosal administration of ETH47 to K18-ACE2 mice prior SARS-CoV-2 infection reduces pathology and increases survival. The manuscript is well written, and the quality of the figures is a real added value. Some of the experiments lack sufficient replicates and adapted statistical analysis.

Main comments

1. Nearly all in vitro experiments are conducted with n=2 while t-test is conducted on sample size < 3, which is not appropriate. t-tests should be conducted after normality distribution confirmation. Authors should add one more experiments and conduct adapted statistical tests.

2. in vivo experiments Fig.4. each dose and time point is representative of only 3 animals and thus rendering statistical comparison difficult; due to natural variations between animals. Adding more animal to critical timepoints appears to be necessary for the interpretation of the data. This should be at least discussed as a strong limitation of the study.

3. Are differences observed between groups in terms of viral replication and symptoms correlated to IFN 1 levels in the lungs?

4. The authors state that mRNA delivery of IFN λ 1 should not allow recipients to develop anti-IFN antibodies. Due to central tolerance, lower type III IFN reactive circulating B and T cells are expected. Peripheral tolerance must be considered since such mRNA platform is widely used to deliver SARS-CoV-2 vaccines. Moreover, in the context of infection, protein overexpression could result in cross-reaction against this protein. It would appear relevant to test at least the absence of any B cell response against hIFN λ 1. This should be at least more discussed.

5. It would have been interesting to assess the dispersion of ETH47 in mice airway to determine which cells produce IFN. This could be discussed.

6. Line 85: infectious animal challenges were conducted with 7.5 ug per animal, without any rationalization of the dose. This dose is not characterized in Fig. 4 and is higher than the previously tested dose (3 ug). This needs further explanation. Any unspecific inflammation could be responsible for the observed effect rather than the induction of ISG. type III IFN are well known inducers of Th1 polarization, which could be an additional mechanism to discuss.

7. One of the most important challenges about SARS-CoV-2 is post viral exposure treatment and immune parameters normalization. It would have been nice that authors tested more precisely the impact of a post-exposure ETH47 treatment in k18-ACE2 mice

Minor comments

8. Except for Fig.3B, all RT-qPCR panels present a dot line at $2^{\Delta\Delta C_t} y=2$, which is confusing. $2^{\Delta\Delta C_t}$ is a fold change; such line should be disposed at $y=1$. Otherwise, this should be stated in the figure legend.

9. Fig.3a indicates that 2ng of mRNA induces more than 100 ng/ml hIFN λ 1 in treated A549 cells, authors could consider including a dose of 200 ng/ml in Fig.3b.

10. Fig.2 caption indicates that $n=2$, though Fig.2C seems to indicate 3 points.

11. Line 162 mentions IAV primers, not used in the study.

Referee #3:

The manuscript by Macht and colleagues investigates a potential use of IFN λ 1-encoding mRNA in the prevention or therapy of SARS-CoV-2. The authors demonstrate that IFN λ 1 mRNA administration results in increased IFN λ 1 protein production in A549 cells, ALI cultures and mouse lungs. This increase in IFN λ 1 levels coincided with increased expression of ISGs (IFIT1, ISG15, OAS3) and decreased SARS-CoV-2 replication in vitro and in vivo. The findings are novel and of broader interest to the scientific community since IFN λ 1-encoding mRNA may also exert protective effects in the context of other respiratory viral infections.

The authors nicely explain their rationale of selecting IFN λ 1 and carefully discuss the advantages of mRNA vs. protein delivery. Another strength is the combination and direct comparison of in vitro (A549, ALI) and in vivo (mice) models. As outlined in more detail below, one major shortcoming is the rather poor description of the experimental setups in the methods section and the figure legends. Furthermore, the number of data points in the figures does not always match the number of experiments/replicates described in the text. These discrepancies need to be addressed.

Major points:

(1) Fig. 2 would benefit from a better description in the legend and a better labeling of the graphs:

- The amount of ETH47 is provided as "ng/well". The authors should also specify the well format that was used or (better) provide the concentration in "ng/ml" as done for recombinant IFN.
- How was recombinant IFN administered in panel B? Was it added to the basal medium and/or the apical surface?
- At what time point were IFN λ 1 levels quantified in the experiment shown in panel B?
- Some data points seem to be missing (e.g. IL-6 quantification at 72 h for 0.05 and 5 ng ETH47). Why?

(2) Fig. 2B: the authors conclude that a significant amount of IFN λ 1 accumulated on the basolateral side. Can the authors rule out that apically secreted IFN λ 1 passed through the cell layer? Did they check integrity/confluence of the monolayer?

(3) Fig. 3 would also benefit from a better description:

- In panel A, the amount of ETH47 is again provided as "ng/well". The authors should specify the well format (as done for panel B) or provide the concentration in "ng/ml" as done for recombinant IFN.
- Which SARS-CoV-2 isolate was used for infection? MUC-IMB-1 is mentioned in line 141, but not listed in the code availability statement.
- The vertical arrow in panel B may be explained in the figure legend itself.
- The description of the ALQ values is not clear to me. Why is the 1 ng/well value problematic if it was above the lower limit of quantification?

(4) Fig. 4B: According to the axis labels, gene expression was calculated as fold change compared to vehicle. Thus, by definition, values of the vehicle control should be 1. However, this does not always seem to be the case, and vehicle controls

(grey) are only shown for some time points. How exactly was the n-fold induction calculated here?

(5) According to line 227 "sufficient material was available for analysis of each sample". However, several data points and time points seem to be missing in Figure 4. Why were these samples not included?

(6) Fig. 5: According to the figure legend, ten mice per group were analyzed in panels A and D. However, only eight and nine data points are shown for the vehicle controls. Why?

Minor points:

(1) Fig. 1 is not essential in my opinion and could be misleading since the present manuscript does not investigate an application of ETH47 in humans.

(2) Lines 27-29: The authors introduce ETH47 as "LNP-formulated modified messenger RNA (mRNA) encoding hIFN λ 1". A better description of the mRNA and LNPs may be helpful if the exact composition is not proprietary (see also line 124).

(3) Lines 52/53: The authors conclude that they observed "no induction of potentially undesired cytokine transcription". However, IL-6 and IL-8 were upregulated up to 7-fold according to Fig. 2D. Thus, this statement should be toned down.

(4) According to the figure legends, "Lipofectamine transfection" was used to deliver LNP-mRNA in the experiment shown in Fig. 3, while cells were "treated" with LNP-mRNA in Fig. 2. How exactly was the mRNA delivered in Fig. 2? Why was lipofectamine transfection required in Fig. 3? Shouldn't the particles enter the cells on their own?

(5) The methods section describes the ELISA used for quantification of IFN λ 1, but not the CXCL9 and CXCL10 ELISAs used in Fig. 4.

(6) Table 1 lists qPCR primers used for the quantification of IAV, although no influenza A virus was used in the present study. Instead, primers used to quantify cytokine expression are missing.

(7) Line 75 mentions IFIT3 although only IFIT1 was analyzed in Fig. 4B.

(8) Lines 188-191 suggest that viral inoculation was performed under sedation of mice with isoflurane gas, while lines 191-193 imply that mice were inoculated under ketamine anesthesia. Which information is correct?

(9) How exactly was infectious virus isolated from mouse lungs (Fig. 5)? A description should be added in the methods section.

ETHRIS GmbH
Dr. Thomas Langenickel
Sommelweisstr. 3
82152 Planegg, Germany
+49 (0)89 89 55 788 0
langenickel@ethris.com

Planegg, 08-May-2024

Dear Reviewers,

thank you for sending us your comments and thoughts, which are very valuable in improving the quality of the manuscript and clarifying statements.

One of the main criticisms is the limited statistical analysis of the data. We addressed the point and implemented a more detailed description of the n-numbers and applied tests in the figure legends and the method section. We agree that a sample size of $n=2$ makes statistical analysis difficult. Nevertheless, the methods we use show high sensitivity and low variances. In *in vivo* experiments we aimed to get a broad view of the mechanistic of IFN λ 1 when using lowest possible number of animals, also in alignment with animal welfare. In the consequent challenge study, we increased animal numbers for statistical comparison, due to expected variability in individuals upon virus challenge. We now address statistical analysis in the discussion of the manuscript.

Furthermore, we have addressed immunological aspects raised by the reviewers by discussing them in the manuscript and added information to the mRNA and LNP design in the method section.

In addition, explanations to the dosages in *in vivo* studies were added and figures and figure legends were revised.

Following, all comments were answered point by point. In the manuscript, the changed parts are marked in the text.

Yours sincerely,

Thomas Langenickel Electronically signed by: Thomas
Langenickel
Reason: I am approving this document
Date: May 8, 2024 18:49 GMT+2

Thomas Langenickel

Referee #1:

Line 5. Remove 'among'.

This was adapted in the manuscript

Line 35. Why only select IFIT1, ISG15, and OAS3 as ISGs to detect?

We thank the reviewer for bringing this point up. Upon activation of STAT-dependent signaling through IFN λ 1 receptor binding, many downstream ISGs are induced. The selected ones were already reported as being highly upregulated in response to IFN λ 1 treatment (Pervolaraki et al.). Furthermore, the investigated targets are specifically known to interfere with virus replication: inhibition of viral infection (IFIT1 and 3), interference with virus production (ISG15), and being key innate immune sensors (OAS1 and 3).

Line 35. The expression of ISGs was only detected by qPCR. The protein level of those ISGs should be determined.

Although we agree with the reviewer that protein detection would be interesting, we originally wanted to prove that treatment with ETH47 succeeds in activating IFN λ 1-pathway via promoting expression of key ISGs. With this in mind, we believe that we achieved the objective of the study and that detecting ISGs at the protein level brings no additional benefit. Since there are no samples left from the study, no additional measurements can be performed.

Line 39-40. The cytokines can induce cytokine release syndrome associated with the severity of the disease. For example, tumor necrosis factor- α causes lethal lung inflammation during SARS-CoV-2 infection. Thus, the level of those cytokines should be investigated.

We thank the reviewer for highlighting this point, and we do envision the suggested investigation in follow-up studies. In this case, we sought to address the expression of key cytokines to provide an initial indication of inflammatory response following ETH47 treatment.

Line 43. Please give the reference about ALI.

We purchased the cultures from Epithelix; the exact reference is already added to the manuscript in material and method section (line 166).

Fig2 a-b. Lack of the result of Control-LNP.

We thank the reviewer for pointing this out. Concerning Fig. 2a, IFN λ 1 concentration in samples treated with Control-LNP was measured for 24h time-point, and values were below limit of quantification. For this reason, we could not include it in the plot. However, we now mention this important point in the main text.

For Fig. 2b, we have added data for Control-LNP accordingly.

Fig2 a, c and d. The statistics should be determined.

We agree with the reviewer. The p-values are now summarized in a table and added as extended view file (Table EV1). Furthermore, we provide data from all doses of the control group (Fig EV1).

Fig2. Why the concentration of Control-LNP was different from ETH47?

After verifying the data, we noticed that the 200 ng/96-well were stated erroneously. The figure should have reported 100 ng/96-well, which is also the highest dose for ETH47. We thank the reviewer for making us aware of this typo.

Fig3 a. The figure should be polished and the statistics should be done between the ETH47 and Control-LNP at different concentrations.

As samples for Control LNP were, as expected, not quantifiable, we could not perform any statistical analysis.

Line 69. As a human protein, how much about the similarity between humans and mice support the activity of hIFN λ in mice?

We thank the reviewer for this important question. In mice, IFN λ 1 is a pseudogene, while IFN- λ 2 and IFN- λ 3 genes encode glycosylated proteins. However, IFN λ R1 shows 99% similarity between mice and humans. Thus, we postulated that human IFN λ 1 should efficiently elicit the expected downstream cascade in mice, which we successfully demonstrated when investigating ISG induction.

Line 75-80. Why the detected mRNA and protein in animals was different from A549 cells?

While we did start to investigate the expression of selected targets in experiments based on cell lines, we opted to extend the selection of targets in animal experiments to account for a much more diversified and complex response, as expected of an in vivo system. Key targets were in any case analyzed in both in vitro and in vivo systems.

Line 85-86. The concentration and times of used ETH47 in animals were different from the former concentration in Figure4, why?

We thank the reviewer for raising this point, for which we need to provide more background in order to answer.

Figure 4 shows a proof-of-concept study performed in-house using single administration of ETH47, without virus challenge. The concentrations in Figure 4 were applied in a total volume of 50 μ L.

Figure 5 represents a virus challenge study, which is why several administrations (pre-treatment only and combination of pre- and post-treatment) were assessed. At the department of Biomedicine at the university of Aarhus, where this challenge study was conducted, only 15 μ L of the test item could be applied. Upon administration of 15 μ L to the nose of the animal, the amount of test item that reaches the lung is lower than for 50 μ L. Therefore, the concentration in the applied solution was increased so that a final dose of 3 μ g in the lung could be reached.

We have now added information in the manuscript to highlight this important point (lines 96-99).

Line92-96. According to Figure, the expression of most ISGs was back to basal level one day post ETH47 treatment. Administering single-dose ETH47 one day before virus inoculation could reduce virus load, how does it work?

We thank the reviewer for drawing attention to this point. As we showed in the previous experiments, ETH47 successfully triggers IFN λ 1 protein production and ISGs induction. Although after 24 hours levels of the analyzed ISGs are decreased, we still reliably detect them above the technical limit of quantification. We speculate that the induced levels of these and additional ISGs, as well as further secondary downstream targets likely to be activated within these 24 hours, are responsible for inducing a primed cellular state conferring a protective barrier when virus infection occurs.

We additionally thank the reviewer for spotting the following aspects.

Line 109. Do you mean that air-liquid interface cultures?

Indeed, we meant what the reviewer rightfully caught; the wording was corrected in the manuscript.

Line 153. Lack all the primers of targeted genes.

TaqMan primers are now included in the method section. We thank the reviewer for noticing.

Line 162. Why does the method contain IAV?

IAV was deleted from the table.

Line 187. How old were those mice used?

We used 6-8 weeks old female mice. This information was added to the manuscript accordingly.

Line 191. Replace 'Sars-CoV-2' with 'SARS-CoV-2'.

We have now changed this as indicated.

Line 217. What is 'DMEM5'?

We thank the reviewer for finding this typo, we meant DMEM.

Line 222. Please give the reference about 'the method of Reed and Muench'.

The reference has been added to the manuscript.

Line 228. In each figure legend, Mann-Whitney's non-parametric U-test or unpaired t test should be detailed.

The used statistical test is now indicated in the figure captions.

Referee #2:

Main comments

1. Nearly all in vitro experiments are conducted with $n=2$ while t-test is conducted on sample size < 3 , which is not appropriate. t-tests should be conducted after normality distribution confirmation. Authors should add one more experiments and conduct adapted statistical tests.

We agree with the reviewer that a sample size of 2 poses difficulties in the statistical interpretation of data. Given the sensitivity of most of the methods we used, and the very low variance, we initially decided that $n=2$ would suffice.

Normality is indeed a pre-requisite of all parametric tests, as rightfully pointed out by the reviewer. However, for very small samples sizes the normality hypothesis can neither be accepted nor be rejected. Moreover, non-parametric tests like Mann-Whitney perform ranking, thus having no power with very small sample sizes regardless of the actual difference of values. For this reason, when statistics was performed a t-test was preferred.

2. in vivo experiments Fig.4. each dose and time point is representative of only 3 animals and thus rendering statistical comparison difficult; due to natural variations between animals. Adding more animal to critical timepoints appears to be necessary for the interpretation of the data. This should be at least discussed as a strong limitation of the study.

We thank the reviewer for addressing once again the importance of sample size. The investigation behind figure 4 was meant to provide a mechanistic view on the effects of ETH47 in vivo, rather aiming to gain a broader view (more time-points and targets) than seeking statistical comparison. For

this reason, we opted for a maximum of 3 animals. An equally important reason for relying on a low number of animals is to align with animal welfare regulations, namely to use the lowest possible number of animals to reach the aim of the study.

However, for the challenge study (figure 5) we focused on statistical comparison and considered data variability. Hence, we analyzed at least 10 animals.

Following the reviewer's indication, we commented on this aspect in the text (lines 79-81).

3. Are differences observed between groups in terms of viral replication and symptoms correlated to IFN λ 1 levels in the lungs?

We did not determine IFN λ 1 protein levels in lungs from in the in vivo virus challenge experiment, but we did detect elevated protein levels in ETH47-treated group up to 24 h post treatment, as shown in the previous experiment (Figure 4), thereby demonstrating expression of the target protein. Moreover, in both vehicle and ETH47-treated animals we would expect additional IFN λ 1 production post infection as native response, which would make the interpretation of such dataset challenging.

4. The authors state that mRNA delivery of IFN λ 1 should not allow recipients to develop anti-IFN antibodies. Due to central tolerance, lower type III IFN reactive circulating B and T cells are expected. Peripheral tolerance must be considered since such mRNA platform is widely used to deliver SARS-CoV-2 vaccines. Moreover, in the context of infection, protein overexpression could result in cross-reaction against this protein. It would appear relevant to test at least the absence of any B cell response against hIFN λ 1. This should be at least more discussed.

We thank the reviewer for the in-depth input. We agree, testing for peripheral tolerance of ETH47 appears to be relevant. However, we see this as a limitation of the study as we only tested single dose administration. The described immune processes would rather be expected to be present upon multiple dose administration. Accordingly, we included an explanation in the discussion (lines 130-133).

5. It would have been interesting to assess the dispersion of ETH47 in mice airway to determine which cells produce IFN. This could be discussed.

We could demonstrate that ETH47-derived IFN λ 1 is a secreted protein which is secreted apically and basolaterally (Fig. 2b) and is efficient in virus neutralization. Therefore, the targeted cell type is not relevant and in our opinion there is no need to investigate this further.

6. Line 85: infectious animal challenges were conducted with 7.5 ug per animal, without any rationalization of the dose. This dose is not characterized in Fig. 4 and is higher than the previously tested dose (3 ug). This needs further explanation. Any unspecific inflammation could be responsible for the observed effect rather than the induction of ISG. type III IFN are well known inducers of Th1 polarization, which could be an additional mechanism to discuss.

The concentrations in Figure 4 were applied in-house in a volume of 50 μ L to the nose. At the department of Biomedicine at the university of Aarhus, where the challenge study (Figure 5) was conducted, only 15 μ L of the test item could be applied to the nose. Upon administration of 15 μ L, the amount of test item that reaches the lung is in proportion lower than for 50 μ L. Therefore, the concentration in the applied solution was increased to be able to deliver a final dose of 3 μ g in the lung. We now commented on that in the manuscript (lines 96-99).

The data in Figure 4c show that no relevant induction of cytokines can be measured at the dose of 3 μ g ETH47, and we assume that also a higher dose, which is considered a small increase in exposure, would not lead to any cytokine increase that would impact study results. Discussing the mechanism of Th1 polarization would go beyond the aim of this study, although we agree with the reviewer that it would be important to address, e.g., in future studies. For now, we mentioned it in the discussion (lines 134-135)

7. One of the most important challenges about SARS-CoV-2 is post viral exposure treatment and immune parameters normalization. It would have been nice that authors tested more precisely the impact of a post-exposure ETH47 treatment in k18-ACE2 mice

Although out of the scope of this specific work, we do agree with the reviewer that this and additional immunological aspects are worth investigating, potentially in more focused follow-up studies. We raised the point now in the manuscript (line 118). We once again thank the reviewer for the valuable input.

Minor comments

8. Except for Fig.3B, all RT-qPCR panels present a dot line at $2^{\Delta\Delta Ct}$ $y=2$, which is confusing. $2^{\Delta\Delta Ct}$ is a fold change; such line should be disposed at $y=1$. Otherwise, this should be stated in the figure legend.

We thank the reviewer for this comment. For robustness, we decided not to consider as target induction any fold change below $y=2$ and indicated this threshold as dotted line at $y=2$. We have specified this information in the figure captions.

9. Fig.3a indicates that 2ng of mRNA induces more than 100 ng/ml hIFN λ 1 in treated A549 cells, authors could consider including a dose of 200 ng/ml in Fig.3b.

The two replicates of cells transfected with 2 ng of mRNA induce a mean concentration of 115 ng/mL IFN λ 1. Thus, we opted for 100 ng/mL, as being more comparable to the mean concentration than 200 ng/mL.

10. Fig.2 caption indicates that $n=2$, though Fig.2C seems to indicate 3 points.

Figure 2c shows no individual data points, but rather mean values. However, we agree that Figure 2b shows 3 individual data points; this is now clarified in the figure caption.

11. Line 162 mentions IAV primers, not used in the study.

We thank the reviewer for noticing, the IAV primers have been deleted from the table.

Referee #3:

Major points:

(1) Fig. 2 would benefit from a better description in the legend and a better labeling of the graphs:
a. The amount of ETH47 is provided as "ng/well". The authors should also specify the well format that was used or (better) provide the concentration in "ng/ml" as done for recombinant IFN.

We thank the reviewer for this input, we have included the well format accordingly.

b. How was recombinant IFN administered in panel B? Was it added to the basal medium and/or the apical surface?

The recombinant protein was added to the apical surface of the insert. This important piece of information is now included in the text.

c. At what time point were IFN λ 1 levels quantified in the experiment shown in panel B?

IFNλ1 mRNA or recombinant protein were removed from the cells 6 h post treatment. IFNλ1 levels were quantified 24 h post treatment. This information was added to the figure caption.

d. Some data points seem to be missing (e.g. IL-6 quantification at 72 h for 0.05 and 5 ng ETH47). Why?

We thank the reviewer for bringing this up. Missing data points indicate that there was no reported Cq value, as this was above the cycle threshold. We added this clarification in the figure caption.

(2) Fig. 2B: the authors conclude that a significant amount of IFNλ1 accumulated on the basolateral side. Can the authors rule out that apically secreted IFNλ1 passed through the cell layer? Did they check integrity/confluence of the monolayer?

We do agree with the reviewer that diffusion of the protein through the cell layer may occur, and indeed we cannot completely rule it out, although checking the integrity and confluency by microscopy. However, diffusion is likely to happen to a similar extent for both the recombinant protein added to the cells and IFNλ1 produced by them. Despite this, we do not observe protein accumulation after treatment with the recombinant protein, suggesting that only ETH47 treatment promotes an active accumulation of IFNλ1 in this compartment.

(3) Fig. 3 would also benefit from a better description:

a. In panel A, the amount of ETH47 is again provided as "ng/well". The authors should specify the well format (as done for panel B) or provide the concentration in "ng/ml" as done for recombinant IFN.

The well format has been included in the labeling in this case, too.

b. Which SARS-CoV-2 isolate was used for infection? MUC-IMB-1 is mentioned in line 141, but not listed in the code availability statement.

We used the SARS-CoV-2 isolate from following study:

<https://www.ebi.ac.uk/ena/browser/view/PRJEB38744>. We have now included the information in the code availability statement.

c. The vertical arrow in panel B may be explained in the figure legend itself.

We have proceeded to explain it as suggested.

d. The description of the ALQ values is not clear to me. Why is the 1 ng/well value problematic if it was above the lower limit of quantification?

*We thank the reviewer for pointing this aspect out. The 1 ng/well sample was analysed undiluted and the measured concentration was above the upper limit of quantification, thereby being indicated as a non-valid value. However, the figure caption erroneously stated: "ALQ = above limit of quantification (sample dilution resulted in a value above LLOQ, real concentration might be slightly different)". We corrected it into: ALQ = above **upper** limit of quantification (sample dilution resulted in a value above **ULOQ**, real concentration might be slightly different).*

(4) Fig. 4B: According to the axis labels, gene expression was calculated as fold change compared to vehicle. Thus, by definition, values of the vehicle control should be 1. However, this does not always seem to be the case, and vehicle controls (grey) are only shown for some time points. How exactly was the n-fold induction calculated here?

The fold change is referred to the mean of the vehicle individual data points. For this reason, values for vehicle itself are slightly scattered around 1.

In the vehicle group, only time-points of 5 and 24 h necropsy were included. The later time-points in the treated group were calculated with reference to the 24 h vehicle time-point.

(5) According to line 227 "sufficient material was available for analysis of each sample". However, several data points and time points seem to be missing in Figure 4. Why were these samples not included?

Similar to question 1d, also here missing values reflect Ct values excluded because higher than the threshold.

(6) Fig. 5: According to the figure legend, ten mice per group were analyzed in panels A and D. However, only eight and nine data points are shown for the vehicle controls. Why?

The missing animals in this groups had to be taken out from the experiment due to biological reasons (e.g. long teeth preventing feeding) before completion of the study or had no usable lung sample due to issues during necropsy. For these mice we are, therefore, missing related data. We thank the reviewer for pointing this out; we have now stated this aspect in the figure caption.

Minor points:

(1) Fig. 1 is not essential in my opinion and could be misleading since the present manuscript does not investigate an application of ETH47 in humans.

While we do agree that we do not provide any evidence in humans, Figure 1 is intended as a general overview of the technology and the rationale behind the ETH47 story, which in turn support the experiments described. We therefore suggest to keep the figure.

(2) Lines 27-29: The authors introduce ETH47 as "LNP-formulated modified messenger RNA (mRNA) encoding hIFN λ 1". A better description of the mRNA and LNPs may be helpful if the exact composition is not proprietary (see also line 124).

We included further information on the mRNA and LNP in the method section (lines 145-156).

(3) Lines 52/53: The authors conclude that they observed "no induction of potentially undesired cytokine transcription". However, IL-6 and IL-8 were upregulated up to 7-fold according to Fig. 2D. Thus, this statement should be toned down.

We agree with the reviewer that, especially prior to what we define plateau (24 hours time-point), we observe IL-6 and IL-8 induction, though mainly with the highest ETH47 dose and not for all replicates (as reflected in the high standard deviation). We adapted the above-mentioned sentence to better reflect this aspect.

(4) According to the figure legends, "Lipofectamine transfection" was used to deliver LNP-mRNA in the experiment shown in Fig. 3, while cells were "treated" with LNP-mRNA in Fig. 2. How exactly was the mRNA delivered in Fig. 2? Why was lipofectamine transfection required in Fig. 3? Shouldn't the particles enter the cells on their own?

As indicated in the title of the plot in Fig. 3a, ETH47 mRNA was applied. By stating "mRNA" we actually refer to the sole transcript without LNP formulation. For this reason, Lipofectamine-mediated transfection was required. On the contrary, cells in experiment shown in Figure 2 were treated with LNP-formulated mRNA, which (as correctly put by the reviewer) enter the cells without any need for further transfection reagents.

(5) The methods section describes the ELISA used for quantification of IFN λ 1, but not the CXCL9 and

CXCL10 ELISAs used in Fig. 4.

We thank the reviewer for noticing it, the methods have been added to the method section.

(6) Table 1 lists qPCR primers used for the quantification of IAV, although no influenza A virus was used in the present study. Instead, primers used to quantify cytokine expression are missing.

Both issues were addressed. IAV primers were deleted and those for cytokine expression included.

(7) Line 75 mentions IFIT3 although only IFIT1 was analyzed in Fig. 4B.

IFIT3 was mentioned by mistake. We have corrected it accordingly in the manuscript.

(8) Lines 188-191 suggest that viral inoculation was performed under sedation of mice with isoflurane gas, while lines 191-193 imply that mice were inoculated under ketamine anesthesia. Which information is correct?

Isoflurane gas was applied for administering the test item, whereas virus inoculation was done under ketamine anesthesia.

(9) How exactly was infectious virus isolated from mouse lungs (Fig. 5)? A description should be added in the methods section.

The reviewer is indeed right, we have missed mentioning this description. We cut lungs into small pieces and further homogenized everything with steel beads in a TissueLyser (II) (both from QIAGEN) in 1ml PBS at 30Hz for 6 min, followed by centrifugation at 300g for 1 min to avoid debris. 10µl of the supernatant were immediately used for TCID50 assay. This information was also added in the method section of the TCID50 assay (lines 233-235).

Dear Dr. Langenickel,

Thank you for the submission of your revised manuscript to our editorial offices. I have now received the reports from the two referees that I asked to re-evaluate the study, you will find below. As you will see, both referees now support the publication of the study in EMBO reports. Both referees have some suggestions to improve the manuscript, I ask you to address in a final revised manuscript. Please also provide a final p-b-p-response regarding the remaining points of the referee.

- Please provide a final title with not more than 100 characters (including spaces).
- We now use CRediT to specify the contributions of each author in the journal submission system. CRediT replaces the author contribution section. Please use the free text box to provide more detailed descriptions and do NOT provide your final manuscript text file with an author contributions section. See also our guide to authors: <https://www.embopress.org/page/journal/14693178/authorguide#authorshippinguidelines>
- Please order the manuscript sections like this, using these names:
Title page - Abstract - Keywords - Introduction - Results & Discussion - Methods - Data availability section - Acknowledgements - Disclosure and Competing Interests Statement - References - Figure legends - Expanded View Figure legends
- The "Data Availability section" should be dedicated to datasets deposited at external repositories. If no such primary datasets have been deposited in any database, please state this in this section (e.g. 'No primary datasets have been generated or deposited') and remove all the other information.
- Please make sure that the number "n" for how many independent experiments were performed, their nature (biological versus technical replicates), the bars and error bars (e.g. SEM, SD) and the test used to calculate p-values is indicated in the respective figure legends (also for potential EV figures and all those in the final Appendix). Please also check that all the p-values are explained in the legend, and that these fit to those shown in the figure. Please provide statistical testing where applicable. Please avoid the phrase 'independent experiment', but clearly state if these were biological or technical replicates. Please also indicate (e.g. with n.s.) if testing was performed, but the differences are not significant. In case n=2, please show the data as separate datapoints without error bars and statistics. See also: <http://www.embopress.org/page/journal/14693178/authorguide#statisticalanalysis>

If n<5, please show single datapoints for diagrams. Presently, some diagrams seem to miss the 'n.s.'. Please check. Moreover:

- Please note that the legends for figures 2b-d is not provided in the sequential manner (legend for figure 2c-d is provided before legend of figure 2b). This needs to be rectified.
- Please note that the exact p values are not provided in the legend of figure 2b.
- Please indicate the statistical test used for data analysis in the legends of figures 5b, e.
- Please note that the error bars are not defined in the legend of figure 5d.
- Please add to each legend (main, and EV figures, where applicable) a 'Data Information' section explaining the statistics used or providing information regarding replicates and scales. See:

- Please make sure that all the funding information is also entered into the online submission system and that it is complete and similar to the one in the acknowledgement section of the manuscript text file. The grant number TPP-2103-0005 is provided in the submission system, but it is missing in the Acknowledgments. Please check.

- Please use our reference format:

- We now request the publication of original source data with the aim of making primary data more accessible and transparent to the reader. Our source data coordinator has already contact you to discuss which figure panels we would need source data for. Please provide the requested source data with your final submission. I attach again the source data checklist. Please upload the completed checklist with your final submission.

In addition, I would need from you:

- a short, two-sentence summary of the manuscript (not more than 35 words).
- two to four short (!) bullet points highlighting the key findings of your study (two lines each).
- a schematic summary figure as separate file that provides a sketch of the major findings (not a data image) in jpeg or tiff format (with the exact width of 550 pixels and a height of not more than 400 pixels) that can be used as a visual synopsis on our

website.

Best,

Referee #2:

The authors provided detailed answers to my questions and clarified their manuscript.
I have one main remaining concern.

1. The authors are right about the power of Mann-Whitney test with small size samples. However, performing T-tests with $n=2$ is not adapted, whatever the sensitivity of the method. The authors should either withdraw statistics or add new sample(s). The missing point to reach $n=3$ only concerns the A549 cell line.
2. line 136 : TH1 should read Th1

Referee #3:

In the revised version of their manuscript, the authors have addressed all of my comments and concerns. More specifically, they have improved the description of the experiments and provided additional details in the figure legends and the methods section. These textual changes have significantly improved the quality of the manuscript. In the legend of Fig. 5, the authors now clarify "that 2 animals of the vehicle group of panel (A) and 1 animal of the vehicle group in panel were excluded from the analysis." For transparency reasons, I encourage them to also provide the reason(s) for excluding these three mice from the analysis. In response to my previous minor comment #8, the authors clarified that "isoflurane gas was applied for administering the test item". I recommend specifying this in line 242 since the sentence following the description of isoflurane sedation mentions both, administration of the test item and virus inoculation.

ETHRIS GmbH
Dr. Thomas Langenickel
Sommelweisstr. 3
82152 Planegg, Germany
+49 (0)89 89 55 788 0
langenickel@ethris.com

Planegg, 11-July-2024

Dear Reviewers,

We kindly thank you for your positive feedback on the revised manuscript.

As further explained in the provided point-by-point reply, and as you can appreciate in the final version of our manuscript, we followed all indicated suggestions.

Moreover, for transparency we would like to bring to your attention additional adjustments that we applied upon checking once again all figures and source data for the final submission as well as journal formatting requirements

In the first round of revision, Reviewer #1 correctly commented on showing 200 ng/96-well for the Control LNP in Fig. 2, whereas the highest ETH47 dose reported was 100 ng/96-well. Though we first identified that erroneously as a typo, the applied dose for the Control group was indeed double the dose of ETH47 treatment. The dose of 200 ng/96-well Control LNP was tested in a dose-range finding study to address potential effects of our LNP on key ISGs. As we show in EV Fig. 1, among these targets only IL-6 was elevated to a minor extent for time points up to 24h. Given these observations, we did not deem it necessary to test lower control doses and reported the highest one in the main figure for reference.

However, to make it more consistent and transparent, we now removed the control group from the main figure and now show all control doses in EV Fig. 1.

In Fig. 4, as well as in the corresponding main text, we are now reporting the administered dose in mg/kg bodyweight, which is easier to interpret than the previously reported estimated dose of deposition in the lung. Moreover, administered doses in Fig. 4 and Fig. 5 now fit to each other – an additional point that was raised during revision.

In Fig. 4C, the unit in the Y-axis was corrected from pg/mL to ng/mL.

In addition, we changed the manuscript title to "*Mucosal IFN λ 1 mRNA-based immunomodulation effectively inhibits SARS-CoV-2 induced mortality in mice*" to fulfill EMBO requirements.

We thank you once again for your excellent review and comments that helped to further improve the manuscript and look forward to a positive response.

Yours sincerely,

Thomas Langenickel Electronically signed by: Thomas
Langenickel
Reason: I am approving this document
Date: Jul 11. 2024 12:57 GMT+2

Thomas Langenickel

Referee #2:

1. The authors are right about the power of Mann-Whitney test with small size samples. However, performing T-tests with $n=2$ is not adapted, whatever the sensitivity of the method. The authors should either withdraw statistics or add new sample(s). The missing point to reach $n=3$ only concerns the A549 cell line.

We agree that statistical analysis based on a sample size of $n=2$ is not appropriate, regardless of the sensitivity and robustness of the test. From a statistical perspective, we additionally envision that adding only one data point would not suffice. In this specific case a proper comparison is not required to convey the message and indeed we added statistics here in reply to another reviewer's request. We removed the statistics in Figure 2 (A, C and D) and Figure 3, as well as statistics shown in expanded view Table 1 (EV1). Furthermore, we now show not only mean \pm SD in Figure 2 and 4, but also the single data points with the mean connecting line.

2. line 136 : TH1 should read Th1

We thank the reviewer for finding this typo, which is now adapted in the manuscript.

Referee #3:

In the legend of Fig. 5, the authors now clarify "that 2 animals of the vehicle group of panel (A) and 1 animal of the vehicle group in panel were excluded from the analysis." For transparency reasons, I encourage them to also provide the reason(s) for excluding these three mice from the analysis.

One animal was excluded due to humane reasons, as being unable to eat due to elongated teeth. For the other animal, processing issues during organ sampling led to high amounts of residual blood in the lung. This circumstance does not support a proper downstream analysis. We included this information now in the figure legend, too. The missing value in Fig. 5D was identified as outlier (ROUT $Q=1\%$) and thus excluded.

In response to my previous minor comment #8, the authors clarified that "isoflurane gas was applied for administering the test item". I recommend specifying this in line 242 since the sentence following the description of isoflurane sedation mentions both, administration of the test item and virus inoculation.

We thank the reviewer for making us aware of that. We deleted "virus inoculum" in the first sentence (Line 244).

Dr. Thomas Langenickel
Ethris GmbH
Germany

Dear Dr. Langenickel,

I am pleased to inform you that your manuscript has been accepted for publication in EMBO reports. Your manuscript will be processed for publication by EMBO Press. It will be copy edited and you will receive page proofs prior to publication. Please note that you will be contacted by Springer Nature Author Services to complete licensing and payment information.

Yours sincerely,
